# The Role of Marine n-3 Polyunsaturated Fatty Acids in Inflammatory-Based Disease: The Case of Rheumatoid Arthritis

**DOI:** 10.3390/md22010017

**Published:** 2023-12-27

**Authors:** Cinzia Parolini

**Affiliations:** Department of Pharmacological and Biomolecular Sciences, Rodolfo Paoletti, Università degli Studi di Milano, Via Balzaretti 9, 20133 Milano, Italy; cinzia.parolini@unimi.it; Fax: +39-02-50318284

**Keywords:** fish oil, inflammation, n-3 PUFAs, rheumatoid arthritis, specialized pro-resolving mediators

## Abstract

Inflammation is a conserved process that involves the activation of immune and non-immune cells aimed at protecting the host from bacteria, viruses, toxins and injury. However, unresolved inflammation and the permanent release of pro-inflammatory mediators are responsible for the promotion of a condition called “low-grade systemic chronic inflammation”, which is characterized by tissue and organ damage, metabolic changes and an increased susceptibility to non-communicable diseases. Several studies have demonstrated that different dietary components may influence modifiable risk factors for diverse chronic human pathologies. Marine n-3 polyunsaturated fatty acids (n-3 PUFAs), mainly eicosapentaenoic (EPA) and docosahexaenoic acid (DHA), are well-recognized anti-inflammatory and immunomodulatory agents that are able to influence many aspects of the inflammatory process. The aim of this article is to review the recent literature that relates to the modulation of human disease, such as rheumatoid arthritis, by n-3 PUFAs.

## 1. Introduction

Chronic non-communicable diseases (NCDs) are still the most common global cause of morbidity and mortality. These illnesses are characterized by the presence of elevated levels of inflammatory markers, mainly cytokines and chemokines at the inflammatory site and in the systemic circulation, and a huge infiltration of inflammatory cells at the site of disease activity [1,2]. NCDs include metabolic syndrome, type 2 diabetes, hypertension, cardiovascular disease (CVD), non-infectious respiratory disease, chronic kidney disease, neurodegenerative and autoimmune diseases (i.e., rheumatoid arthritis), depression, osteoporosis, age-related macular degeneration and various types of cancers [2,3].

Inflammation is a conserved process that involves the activation of immune and non-immune cells aimed at protecting the host from bacteria, viruses, toxins and injury, and, consequently, removing the pathogens and supporting tissue repair and recovery [4,5]. Within minutes of recognizing a harmful signal, the acute inflammatory response starts with an “onset phase” that involves the production of chemokines, cytokines, eicosanoids, proteases, vasoactive amines, neuropeptides and neurotransmitters by resident immune and structural cells. Moreover, this process is characterized by the recruitment of different cell types, such as granulocytes, from blood to the tissue inflammatory site [6]. In physiological conditions, the inflammatory response is programmed to evolve in an active “resolution phase” that is characterized by highly coordinated cellular and molecular events, including the release of anti-inflammatory cytokines and pro-resolving mediators, the loss of receptors for pro-inflammatory stimuli and the activation of regulatory cells that weaken the activity of pro-inflammatory cells [7]. The effectiveness of the “resolution phase” is conditional upon specific cellular mechanisms that are orchestrated by pro-resolving mediators engaged in halting the inflammatory response and initiating tissue repair and healing. Specifically, the recruitment of non-phlogistic monocytes and their differentiation into macrophages able to clear the local leukocytes by apoptosis and subsequent phagocytosis are some of the key events that determine the initiation of the “resolution phase” [8]. In addition, the apoptotic cells “inform” the phagocytosing macrophages that the inflammatory response is ending, which triggers the macrophage mediator production to switch from a pro-inflammatory (M1) to an anti-inflammatory and pro-resolving phenotype (M2) [9]. It has been observed that the balance between the M1 and M2 macrophages is fundamental for the proper resolution of inflammation [10]. Others cellular actors of the “resolution phase” are (1) regulatory T cells (Tregs), which release anti-inflammatory cytokines, such as interleukin 10 (IL-10) and transforming growth factor-beta (TGF-beta), and remove IL-2, which is essential for the activation of T cells [11]; (2) innate lymphoid cells (ILCs), such as ILC2, ILC3 and NK cells [12]; (3) myeloid-derived suppressor cells (MDSCs), which possess immunomodulatory activities through the induction of Treg cell expansion and the production of IL-10 and TGF-beta. Moreover, the MDSCs mediate the efferocytosis of apoptotic neutrophils [13,14]. Meanwhile, the lipid mediator class switching promotes a shift from the synthesis of prostaglandins (PGs) and leukotrienes (LTs) via 5-lipoxygenase (LOX) in inflammatory exudates to the production of lipoxin A4 via 15-LOX and the following reprogramming of granulocytes to initiate the “resolution phase” [15]. As stated above, during the “resolution phase”, several mediators are produced to prevent the exacerbation of acute inflammatory mechanisms and eventually restore tissue homeostasis. These anti-inflammatory and pro-resolving mediators display peculiar activities: (i) halting or inhibiting neutrophil recruitment; (ii) promoting the influx of non-phlogistic monocytes; (iii) inducing neutrophil apoptosis and efferocytosis by macrophages; (iv) promoting the shift from M1 to M2 macrophage phenotypes; (v) organizing the return of non-apoptotic cells to the blood or egress via the lymphatic vasculature; (vi) stimulating the cellular repopulation of the tissue, aimed at adaptive homeostasis [16].

Nevertheless, unresolved inflammation and the permanent release of pro-inflammatory mediators are responsible for the promotion of a condition called “low-grade systemic chronic inflammation”, which is characterized by tissue and organ damage, metabolic changes and an increased predisposition to NCDs [2,17]. In the last twenty years, the therapeutic approach has been focused on the possibility of positively impacting the “resolution phase” of the inflammatory process [18].

Rheumatoid arthritis (RA) is an autoimmune disease that leads to progressive joint damage, threatening the quality of life and increasing functional disability. The pathogenesis of RA is unknown, but starting from the beginning of the 21st century, dramatic improvements in understanding the basic mechanisms have been made, leading to significant changes in RA therapies [19,20]. Additionally, diverse studies have demonstrated that different dietary components may influence modifiable risk factors for diverse chronic human pathologies [21]. Particular attention has been drawn to fish consumption since it lowers plasma lipid concentrations and attenuates inflammation. Indeed, fish is the main source of omega-3 long-chain polyunsaturated fatty acids (n-3 PUFAs) [22].

The aim of this article is to review the recent literature that relates to the modulation of inflammatory-based disease, such as RA, by n-3 PUFAs.

## 2. n-3 Polyunsaturated Fatty Acids (n-3 PUFAs) and Specialized Pro-Resolving Mediators (SPMs)

n-3 fatty acids are a family of n-3 PUFAs characterized by the presence of a final double bond between carbons 3 and 4. The main components of this family are eicosapentaenoic (EPA), docosapentaenoic (DPA), and docosahexaenoic (DHA) acid. EPA and DHA are abundant in the flesh of both lean and oily fish (with the greater amount being for DHA) and in supplements, such as fish oils, cod liver oil and krill oil, and in some algal oils [23]. It is widely recognized that the anti-inflammatory activity of n-3 PUFAs involves their incorporation into cell membrane phospholipids at the expense of arachidonic acid (ARA). This effect causes a decrease in the amount of AA in the membranes that inhibits ARA metabolism and expression of the cyclooxygenase (COX) gene, eventually decreasing the production of ARA-derived eicosanoids [24]. These processes are triggered by inflammatory stimuli, which activate the phospholipase A2 enzyme that is responsible for the hydrolysis of the sn-2 acyl chain of glycerol phospholipids, generating free AA or free EPA or DHA. Indeed, AA, an n-6 PUFA, is found to be esterified at the sn-2 position of membrane phospholipids [25]. Three main enzymatic pathways are responsible for eicosanoid production from ARA: (1) COX (COX1 and COX2); (2) 5-LOX, 12-LOX and 15-LOX; and (3) cytochrome 450 mixed-function oxidase enzymes (CYP450). The classical eicosanoids are PGs, thromboxanes (TXs) and LTs, which are the best-known mediators and regulators of inflammation [26]. Of note, human beings evolved on a diet in which the ratio of n-6/n-3 PUFAs was about 1, whereas in Western diets, this ratio is 15.9/1. This change upset the metabolic balance that was characteristic during evolution, when our genes were programmed to respond to a different type of diet [27]. n-6 PUFA metabolites are pro-inflammatory and pro-atherothrombotic molecules, which play a key role in allergic reactions and inflammatory disorders, as well as in the metabolic pathways regulating appetite and food intake that eventually lead to an increase in body weight/obesity [28]. On the other hand, n-3 PUFAs (such as EPA and DHA) possess anti-inflammatory, anti-aggregatory, vasodilation and bronchodilation effects [28]. Diverse studies have demonstrated a higher activity of EPA compared with DHA [24]. However, no consistent data on this regard have been produced up to now [24].

Several in vitro and in vivo studies demonstrated that n-3 PUFAs decrease the cell-surface expression of adhesion molecules and the production of inflammatory cytokines (such as tumor necrosis factor (TNF)-alpha, IL-1 beta and IL-6) and COX-2 metabolites [29]. These effects are mediated by the ability of n-3 PUFAs to impact the nuclear factor kappa B (NFkB) pathway [23]. Specifically, NFkB is one of the transcription factors engaged in the upregulation of genes encoding inflammatory proteins. The cytosolic and inactive form of NFkB is a trimer, and its activation is promoted by inflammatory stimuli through a signaling cascade that includes endotoxins (for example, lipopolysaccharides, LPS) that bind to toll-like receptor (TLR) 4. This cascade starts with the phosphorylation of the inhibitory subunit of NFkB, which then dissociates from the trimer; the remaining dimer is able to translocate into the nucleus, where it binds to response elements and upregulates inflammatory genes [30,31]. Currently, three alternative mechanisms have been proposed to explain the ability of n-3 PUFAs to dampen inflammatory signaling via NFkB: (1) activation of peroxisome proliferator-activated receptor (PPAR)-gamma, which physically interacts with the dimeric form of NFkB, preventing its nuclear translocation; (2) interfering with raft formation in the membrane of inflammatory cells via TLR 4 and myeloid differentiation primary response gene 88 (MyD88); (3) binding to G-protein-coupled cell-membrane receptor (GPR120), which triggers an anti-inflammatory cascade stimuli that is able to inhibit NFkB activation [23].

Besides exerting these anti-inflammatory activities, n-3 PUFAs have been implicated as substrates for the synthesis of specialized pro-resolving lipid mediators (SPMs). These SPMs are molecules recognized as having a central role in inflammation resolution and in the protection of the organism against the harmful consequences of uncontrolled inflammatory process [32]. SPMs, synthesized by the action of COX, LOX and CYP450 enzymes, include resolvins (Rvs), protectins (a.k.a. neuroprotectins, PD1/NPD1), maresins (MaRs), and the novel cysteinyl-SPMs (cys-SPMs). In addition, in the presence of aspirin, different epimers of SPMs are produced and are called aspirin-triggered (AT) SPMs [7]. These biosynthetic pathways may occur within a single cell type (if all the enzymes are expressed) or in a transcellular manner, with the early steps occurring in one cell type and the following once in a different cell that synthetizes the final biologically active lipid mediator [33]. Specifically, SPMs play a key role in several pro-resolving mechanisms: (1) limiting granulocyte chemotaxis and infiltration in vivo; (2) stimulating macrophage phagocytosis and efferocytosis; (3) enhancing macrophage M2 polarization; (4) accelerating wound healing; (5) reducing the production of pro-inflammatory cytokines (TNF-alpha and IL-1beta) and lipid mediators (PGs and LTs); (6) promoting the Treg response and release of IL-10; (7) reducing platelet aggregation and inflammasome formation [7]. Moreover, studies performed in animal models of both acute and chronic inflammation have reported the therapeutic efficacy of SPMs [33]. These data could open a new strategy for the treatment of diverse inflammatory-based pathologies, being, at least in part, free of the adverse effects commonly observed in clinical trials using standard-of-care therapies [34].

DHA is the precursor of D-series Rvs, PD1/NPD1, MaR1 and cys-SPMs (Figure 1).

Studies have demonstrated that RvDs are strong immunoresolvent agents that act mainly through two G-coupled receptors, ALX/DRV1/GPR32 [35] and DRV2/GPR18, which bind specifically to RvD1 and RvD2, respectively [36]. PD1/NPD1, through its receptor GPR37, exerts its anti-inflammatory functions in neural (NPD1) and immune systems (PD1). Studies have demonstrated its ability to protect the host from ischemic stroke, retina degenerative disease and traumatic brain injury [37]. Structural studies have identified a positional isomer of PD1, called PDX, that is able to inhibit platelet activation, enhance insulin sensitivity and reduce atherosclerosis, even though a specific receptor still needs to be identified [38]. MaR1, a DHA-macrophage derived maresin that is synthetized by 12-LOX pathways promotes the fundamental pro-resolving functions of macrophages by interacting with a cell-surface leucin-rich repeat-containing G-protein-coupled receptor, LGR6 [39]. Cys-SPMs are a group of pro-resolving and pro-repair mediators containing three series of peptide–lipid conjugated SPMs, such as the resolving conjugates in tissue regeneration (RCTRs) [40], protectin conjugates in tissue regeneration (PCTRs) [41], and maresin conjugates in tissue regeneration (MCTRs) [42].

E-series Rvs (RvE1, RvE2, RvE3 and RvE4) are, instead, biosynthesized from EPA, and RvE1 and RvE2 act by interacting with their receptors, ERV1/ChemR23 and BLT1, respectively (Figure 2A) [43]. RvE1 binding to ERV1/ChemR23 promotes the activation of intracellular signals, such as kinase phosphorylation. Experimental studies using agonists to the RVE1 receptor highlighted that the stimulation of the endogenous resolution mechanisms may control both inflammation and cancer progression [37].

Alongside the above cited and well-characterized DHA- and EPA-derived SPMs, almost ten years ago, Dalli et al. identified analogous compounds synthetized from the third marine n-3 PUFAs, i.e., DPA (Figure 2B), that have been assigned to the Rv, PD and MaR families [44]. Specifically, they were designated as RvDn-3 DPA (RvD1,2,5n-3 DPA), PDn-3 DPA (PD1,2,3n-3 DPA) and MaRn-3 DPA (MaR1,2,3n-3 DPA) (Figure 2) [44]. In addition, a few years later, the same authors characterized the structure and function of novel 13-series resolvins, named RvT1, RvT2, RvT3 and RvT4, specifically produced in the circulation during the interaction between vascular endothelial cells and neutrophils (Figure 2B) [45]. Interestingly, the synthesis of these new immunomodulatory compounds is stimulated by statins, particularly atorvastatin and pravastatin, via the nitrosylation of COX-2, which enhances its enzymatic activity. A mouse model of arthritis has been used to demonstrate that both atorvastatin and pravastatin increase the tissue and plasma concentrations of RvTs and are able to diminish the joint disease as well as leukocyte trafficking and activation [46].

## 3. Rheumatoid Arthritis (RA)

RA is a systemic autoimmune disorder that is characterized by chronic inflammation of the joints, manifested as swelling, pain, functional impairment and morning stiffness, and the production of autoantibodies, i.e., rheumatoid factor (a high-affinity autoantibody against the Fc portion of immunoglobulin) and anti-citrullinated protein antibody (ACPA) [47]. Additionally, RA includes systemic features, such as cardiovascular disease (CVD), and pulmonary, psychological, and skeletal disorders [48,49,50]. Any peripheral joint can be affected, but the most commonly affected joints are within the feet, knees and hands. Of note, other parts of the body can be a target of this disease, such as the skin, eyes, lungs, heart, nerves and blood [49,50]. The prevalence of RA ranges from 0.5% to 1% of the population worldwide, with the predominant percentage being women [51]. The etiological agent responsible for triggering the onset of RA is unknown. However, it is well recognized that RA comprises a complex interplay among genotype, environmental triggers, and change [48]. Conventional and genome-wide approaches have been applied to identify more than 100 loci associated with the RA risk and progression [52,53]. These studies have helped to better clarify the contribution to RA pathogenesis of fundamental genes, pathways and cell types. Additionally, they have supplied experimental evidence that the genetics of RA could drive drug discovery [52,53]. Specifically, the majority of these loci are genes encoding (i) MHC class II molecules, primarily the HLADR locus, which is implicated in the T-cell recognition of autoantigens [54]; (ii) co-stimulatory pathways (CD28, CD40, chemokines and cytokine receptors) [55], post-translational modification enzymes (i.e., peptidyl arginine deiminase, type IV (PADI4)), responsible for the conversion of peptidylarginine to citrulline] [56], and intracellular regulatory pathways (i.e., PTPN22, TNFAIP3, STAT3) [57,58], all of which are potentially able to modify the threshold for immune activation or failed regulation. These observations have been confirmed by studies of gene–environment interactions. Indeed, smoking and other forms of bronchial stress (e.g., exposure to silica dust) are directly associated with an increase in the RA risk among persons with susceptibility HLA-DR4 alleles [59,60], while smoking and HLA-DRB1 alleles work together in raising one’s risk of having ACPA [61]. Additionally, environmental stressors of pulmonary and other barrier tissues could trigger the post-translational activity of PADI4, eventually leading to quantitative or qualitative alterations in the citrullination of mucosal proteins [62]. Among the citrullinate self-proteins recognized through the diagnostic assay are alpha-enolase, keratin, fibrinogen, fibronectin, collagen and vimentin [63]. Infection diseases caused by diverse virus and bacteria species, as well as by their products, have been correlated with the onset of RA [64]. The formation of immune complexes during infection may indeed stimulate the synthesis of rheumatoid factors [65]. Since it has been observed that *Porphyromonas gingivalis* expresses PADI4, RA seems to be associated with periodontal disease [66]. Lastly, the gastrointestinal microbiota has been identified as a factor influencing the development of autoimmunity, because it is involved in maintaining immune homeostasis and acts as a marker of the host’s health status [67,68]. Perturbation of this interaction can affect mucosal and systemic immunity and promote diverse inflammatory and autoimmune diseases such as RA [69]. Many critical questions still need an answer; primarily, why does the systemic loss of tolerance transform a pauci-cellular synovium into chronically inflamed tissue? Nonetheless, progress in understanding the pathogenesis of the disease has promoted the discovery of new pharmacological treatments, leading to improved outcomes [48].

It is conceivable that infections or traumatic insults could induce the initial inflammatory response in the synovium, causing synovitis. Synovium is the tissue lining the joints, and it is the place where the infiltration of neutrophils, B and T lymphocytes and monocyte-derived macrophages and the proliferation of fibroblast-like synovial cells, called synoviocytes, occurs [70]. Of note, a multicenter, randomized, double-blind, controlled study demonstrated that the selective depletion of B cells through the administration of rituximab, a genetically engineered chimeric anti-CD20 monoclonal antibody, to patients with active RA, proved efficacious in ameliorating disease symptoms [71]. As the disease develops, the cells of the synovial lining proliferate, forming an invasive pannus together with new blood vessels, which leads to the progressive destruction of cartilage and bone [72]. This condition can lead to a multifactorial syndrome named osteosarcopenia, which is characterized by muscle wasting and an increased risk of osteoporosis [73]. Synovial B cells are mainly localized in T-cell–B-cell aggregates, and their activation is endorsed by the expression of factors, including a proliferation-inducing ligand (APRIL), B-lymphocyte stimulator (BLyS) and CC and CXC chemokines [74]. Importantly, macrophages play a key role in synovitis. Indeed, clinically effective biological agents aim to hamper macrophage infiltration in the synovium [75]. Specifically, by looking at the pattern of pro-inflammatory cytokine expression, the hypothesis is that the M1 macrophage phenotype is the predominant one [48]. There are diverse macrophage-activating pathways primarily involving (i) toll-like receptors (TLRs) 2/6, 3, 4 and 8; (ii) nucleotide-associated molecular patterns; (iii) damage-associated molecular patterns, such as bacterial, viral and putative endogenous ligands; (iv) cytokines; (v) cognate interactions with T cells; (vi) immune complexes; (vii) lipoprotein particles; (viii) liver X-receptor agonists, such as oxysterols and oxidized low-density lipoproteins (LDL); (ix) serum amyloid A-rich high-density lipoproteins (HDL); and (x) protease-rich microenvironments through protease-activated receptor 2 [76]. High concentrations of different cytokines and chemokines, including IL-1beta, IL-6, IL-8, IL-21, IL-23, IL-17A, IL-17F and TNF-alpha, have been detected in the synovial fluid of patients with RA and have been associated with the development of the disease [77]. Of note, TNF-alpha and IL-6 play a fundamental role in RA, which was confirmed by the successful therapeutic blockade of membrane and soluble TNF-alpha and the IL-6 receptor in patients affected by RA [48]. Indeed, TNF-alpha triggers the expression of cytokines, chemokines and endothelial-cell adhesion molecules; protects synovial fibroblasts; promotes angiogenesis; suppresses regulatory T cells; and induces pain sensation [78,79]. Meanwhile, IL-6 orchestrates the local leukocyte activation and autoantibody production together with systemic effects that promote acute-phase responses, anemia, cognitive disfunction, and lipid-metabolism dysregulation [80]. An additional pathogenic pathway includes the antigen-nonspecific, T-cell-contact-mediated activation of macrophages and fibroblasts, acting through their ligand’s interactions, i.e., CD40-CD40L, Cd200-CD200L, intracellular adhesion molecule 1 (ICAM-1) and leukocyte-function-associated antigen 1 [81]. Moreover, the AA-derived lipid mediators of inflammation, PGE2, LTB4, 5-hydroxyeicosatretraenic acid and platelet-activating factor (PAF), are also found in the synovial fluid and are believed to be involved in the recruitment of leukocytes into the diseased tissue [82]. Finally, as stated above, RA is characterized by bone erosion that occurs rapidly (affecting 80% of patients within 1 year after diagnosis [83]) and is correlated with prolonged, increased inflammation [84]. Synoviocytes, a.k.a. fibroblast-like synovial cells, within the synovia lining layer are the major source of matrix metalloproteinases (MMPs), such as MMP1, 3, 9, 10 and 13, which accelerate bone erosion in the joints [85]. The secretion of these MMPs is stimulated by the TGF-beta signaling pathway (also called the bone morphogenic protein (BMP) pathway) through the site-specific phosphorylation of SMAD proteins [86]. Synovial cytokines, specifically macrophage colony-stimulating factor (M-CSF) and receptor activator of NF-kB ligands (RANKL), promote osteoclast differentiation and migration of the periosteal surface adjacent to articular cartilage [87]. Importantly, RANKL, a member of the TNF superfamily, exerts its activity by interacting with its receptor, RANK. Osteoprotegrin (OPG) is a soluble decoy receptor for RANKL and inhibits RANKL functions by competing with the binding between RANK and RANKL [88]. The activation of this osteoclast signaling pathway leads to the expression of osteoclast-specific proteins that are necessary for demolishing mineralized tissues, including mineralized cartilage and subchondral bone. This acidic activity of osteoclasts leads to deep resorption pits, which are replaced by inflammatory tissue [87]. Clinical studies have confirmed the pivotal role of these cytokines because their inhibition halts erosion in RA [89].

The clinical diagnosis of RA is based on criteria developed by the European and American Rheumatology Association [90] and treatments are focused on disease-modifying antirheumatic drugs (DMARDs) that target remission of the disease [90]. Several studies have shown that starting the treatment with DMARDs at the early stages of RA is associated with a positive change in the progression of the disease, leading to a slowdown in joint destruction [91]. The number of therapeutic options available for the treatment of RA has increased dramatically in the past 30 years [92]. Specifically, non-steroidal anti-inflammatory drugs, glucocorticoids [93], and DMARDs of synthetic origin (i.e., methotrexate or janus kinase (JAK)-inhibitors) or of biological origin (i.e., TNF inhibitors, costimulation modifiers, IL-6 inhibitors and B-cell-depleting drugs) are included in the first-line protocols for RA patients [94,95]. Additionally, validated composite disease measures, standardized strategies such as “treat to target”, and the stringent definition of remission could help to reach the aim of clinical remission, at least in early RA [96]. Furthermore, starting from the drugs’ mechanism of action, it has been possible to focus on several biomarkers that serve as indicators of organ activity, pathogenic processes or response to treatment [97]. Several types of biomarkers have been identified and utilized for the (i) diagnosis of disease; (ii) prediction and assessment of prognosis; (iii) prediction and assessment of response to therapy; and (iv) prediction and assessment of drug toxicity [98]. Nevertheless, due to the heterogeneity of RA pathophysiology, many patients do not attain clinical remission, resulting in reduced mobility, disability, and low quality of life [99]. Additionally, lifestyle changes (i.e., smoking cessation, dental care, weight control), assessment of vaccination status, and management of comorbidities should be included in standard-of-care therapies [100].

Among the modifiable factors influencing the development and the progression of RA, smoking, alcohol and unsaturated fatty acid (FA) consumption have been implicated as the most important predictors [101]. For this reason, even if the inflammation can be controlled by biologic agents, appropriate nutritional intervention should still be considered for these patients. In this context, nutrition plays both direct roles in disease development, by carrying pro- or anti-inflammatory food components, and indirect roles, through effects on the body mass index, visceral fat accumulation and by contributing to the onset or the prevention of diabetes and CVD [102]. In addition, changes in lifestyle, of which healthy eating is one pillar, physical activity, management of stress and avoiding risky substances and social connections may be effective in preventing RA. The most popular nutritional approaches include the Mediterranean diet and dietary supplements, especially fish oil or n-3 PUFAs [103].

Importantly, n-3 PUFAs and their SPM lipid mediators (Figure 3), recognized as anti-inflammatory molecules through their ability to suppress inflammatory signaling via NFkB [20], have been widely investigated in the context of RA. Two experimental studies showed that n-3 PUFAs, i.e., EPA and DHA, reduced the incidence and severity of arthritis in two different animal models of arthritis [20,104,105]. Additionally, these results highlighted that EPA was more effective than DHA [104,105]. Furthermore, Ierna et al. [106] demonstrated that in mice subjected to collagen-induced arthritis, the administration of krill oil, which provides marine n-3 PUFAs partly in the form of phospholipids, was more efficacious than fish oil, mainly composed of n-3 PUFAs in triacylglycerol form, in slowing the onset of arthritis, reducing its severity, decreasing paw swelling, and reducing knee-joint pathology. These results are in line with studies showing a more effective delivery of marine n-3 PUFAs from phospholipids than from triacylglycerols [107].

The dietary consumption of n-3 PUFAs could play a key role in the development of RA. An observational study by Di Giuseppe et al. found that a dietary n-3 PUFAs intake greater than 0.21 g/day was associated with a 35% lower risk of developing RA compared with women consuming a reduced amount [108]. Moreover, long-term consumption of fish >1 serving/week was correlated with a 29% reduction in RA risk compared with <1 serving/week [108]. Similar results were reported by Rosell [109] and Pedersen [110], confirming the association between n-3 PUFA consumption and a reduction in the development of RA. In addition, a recent systematic review and meta-analysis, which included seven cohort and six case-control studies, demonstrated that for every 100 g/day increment in fish intake, there was a 15% lower risk of developing RA [111].

Of note, nutrition has been shown to have an influence on disease flares, overall management, and clinical outcomes. One of the first studies was conducted by Kremer et al. [112]. RA patients were treated with 1.8 g of EPA for 12 weeks. At the end of the treatment, patients manifested both a reduction in (1) joint stiffness in the morning and (2) the number of tender joints. The plasma concentration of IL-1 beta was significantly reduced (*p* < 0.001) in RA patients treated with 3.6 g/day of n-3 PUFAs for 12 weeks [113]. Additionally, fish oil supplementation proved to ameliorate the RA disease [114]. Specifically, this study demonstrated a reduction in the AA/EPA ratio as well as the mean LTB4 production by 30% and 33%, respectively, together with a drop (minus 37%) in PAF production [114]. These data were later confirmed by Dawczynski [115]. Of note, this study also detected (i) a significant decrease in the erythrocyte membranes’ AA/EPA ratio; and (ii) an immune response inhibition through the reduction in the number of lymphocytes and monocytes recruited [116]. Diverse studies revealed that patients taking n-3 PUFAs achieved significant improvements in global assessment and pain and a reduction in antirheumatic medication (about 30%) [117,118,119]. In addition, in the study by Geusens et al. [118], it was observed that a high dose (2.6 g/day) proved more efficacious in reducing RA pain than a low dose (1.3 g/day). Meta-analyses have been published to provide information concerning the influence of n-3 PUFAs on the clinical manifestation in RA patients. Goldberg et al. [120] evaluated the pain-relieving effects of EPA/DHA in RA patients. Seventeen randomized controlled trials (RCTs) were included in the final meta-analysis. Patients receiving different doses of n-3 PUFAs, ranging from 1.8 to 9.6 g/day, were compared with individuals receiving different types of placebos (such as soy oil, linoleic acid capsules, air-filled capsules, olive oil, coconut oil, corn oil, and water). Six different pain outcomes were identified: (1) patient-assessed pain; (2) physician-assessed pain; (3) duration of morning stiffness; (4) number of painful and/or tender joints; (5) the Ritchie articular index [121]; (6) nonselective nonsteroidal anti-inflammatory drug (NSAID) consumption. The results of this meta-analysis suggest that n-3 PUFA supplementation reduces patient-assessed pain (−0.26; 95% CI: −0.49 to −0.03; *p* = 0.03), the number of minutes of morning stiffness (−0.43: 95% CI: −0.72 to −0.15; *p* = 0.003), the number of painful and/or tender joints (−0.29; 95% CI: −0.48 to −0.10; *p* = 0.003), and NSAID consumption (−0.40; 95% CI: −0.72 to −0.08; *p* = 0.01). However, no significant effects were observed for physician-assessed pain (−0.14; 95% CI: −0.49 to 0.22; *p* = 0.45) or the Ritchie articular index (0.15; 95% CI: −0.19 to 0.49; *p* = 0.40) [120]. In order to maximize the therapeutic efficacy, the authors suggested that for future trials, (i) all studies should clearly state the dose and type of NSAIDs used, and (ii) a non-olive oil placebo should be included as a control. The reason is based on the anti-inflammatory proprieties of olive oil, as previously demonstrated. Indeed, its main constituent, oleic acid, may compete with AA for incorporation into phospholipids, while minor components, such as tyrosol and beta-sitosterol, possess anti-oxidative and anti-inflammatory effects [120,122]. In 2012, Lee et al. published a meta-analysis considering only the ten RCTs, already included in the previous study, in which RA patients were treated with a dose of n-3 PUFAs above 2.7 g/day for a minimum duration of three months [123]. No clinical outcome measures were improved by n-3 PUFA supplementation, with only a trend toward an improved tender joint count, swollen joint count, morning stiffness, and physical function recovery observed in the n-3 PUFA-treated group. The authors highlighted that NSAID requirements were significantly different between the n-3 PUFA and the placebo groups. However, this association was gathered from only two case-control studies. Among the outcomes evaluated in this meta-analysis, no results were reported concerning the impact of n-3 PUFAs on the erythrocyte sedimentation rate (ESR) and C-reactive protein (CRP) [124]. A more recent meta-analysis was published by Gioxari [51]. This meta-analysis included 20 RCTs with the following inclusion criteria: written in English; conducted in humans; oral intake of n-3 PUFAs (either as a supplement or from food source); duration of the study greater or equal to three months; maintenance of conventional drug treatment for the entire course of the study. The outcomes evaluated were: (i) changes in RA disease activity (through the monitoring of 16 RA markers); (ii) inflammation (measuring CRP, IL-1, IL-6, TNF-alpha and LTB4); (iii) risk for CVD (assessing CVD-related risk factors, such as TG, TC, LDL and HDL). Oral treatment with n-3 PUFAs proved efficacious in ameliorating several RA disease markers, namely early morning stiffness, tender joint count, the pain scale, the Ritchie articular index, improving the results from the health assessment questionnaire, grip strength, ESR, and reducing plasma levels of LTB4 and TG. No effects were detected on TC, LDL and HDL plasma concentrations [51]. These data were also confirmed by a randomized, single-blind intervention study. RA patients were treated with 1800 mg/day of EPA and 1200 mg/day of DHA for 90 days. Compared with the baseline values, RA patients showed reductions in diverse RA disease markers and CRP levels [124]. Additionally, an in vitro study found that Rvd5 suppressed Th17 cell differentiation and CD4+ T cell proliferation, facilitated Treg differentiation, and inhibited osteoclastogenesis [125]. Of note, a cross-sectional study demonstrated that taking oral over-the-counter fish oil supplementation increased DHA and EPA plasma concentrations as well as SPM precursors levels. Specifically, they detected a rise in the concentration of both 18-hydroxy-EPA and 17-hydroxy-DHA, which have been proved to be potent inhibitors of macrophage-mediated pro-inflammation in addition to being substrates for SPMs [126]. Finally, two systematic reviews highlighted, on the one hand, an inverse correlation between the highest vs. lowest category of fish consumption and RA risk, and on the other hand, the efficacy of a dose > 2 g/day of PUFAs in improving tender and swollen joints, morning stiffness, and a reduction on NSAID use [127]. Meanwhile, a meta-analysis of 23 randomized placebo-controlled trials revealed only a trend toward beneficial effects following n-3 PUFA supplementation in improving RA disease parameters [128]. The strengths of this meta-analysis are: (1) relatively large number of retrieved trials; (2) only placebo-control clinical studies and patients with defined diagnostic criteria were included; (3) the choice of appropriate effect estimates wherever possible [128].

## 4. Conclusions and Perspectives

Inflammation is a physiological process within the immune response that is involved in protecting the organism against pathogens, in the reaction to injury and in wound healing. The inflammatory response is normally a self-limiting and resolving process, thanks to the activation of diverse negative feedback mechanisms. Indeed, in parallel with an appropriate functioning of the inflammatory response are numerous pro-resolving lipid mediators, such as n-3 PUFA-derived SPMs, that are implicated in the active “resolution phase” of the inflammation. However, excessive or uncontrolled inflammation can be harmful and responsible for tissue damage and for the pathogenesis of several human disease. Marine-derived n-3 PUFAs have shown potential efficacy in reducing inflammation and promoting resolution, thus preventing or modulating several NCDs, such as rheumatoid arthritis. Altogether the data reported in this review show that anti-inflammatory interventions, i.e., high fish consumption or supplements containing n-3 PUFAs, should be the standard of care, along with pharmacotherapy, in treating patients with RA. However, all the investigators emphasize the need for longer, larger and well-designed clinical studies to reach a final conclusion. It is important to underline that researchers may need to focus on the diversity of lipid-derived mediator classes to better understand the therapeutic potential of n-3 PUFA supplements. Finally, new biomarkers of the RA disease have been identified thanks to omics approaches, which are those coming from the investigation of lipids (lipidomics), proteins (proteomics) and metabolites (metabolomics) [129]. Specifically, clinical lipidomics, by allowing a detailed lipidome profiling of plasma/serum/lipoproteins and immune cells of RA patients, could trigger the identification of so-called “personalized medicine”, together with an improvement in the early and precise diagnosis, management of disease progression, and evaluation of new drug strategies and outcomes [130].

## Figures and Tables

**Figure 1 marinedrugs-22-00017-f001:**
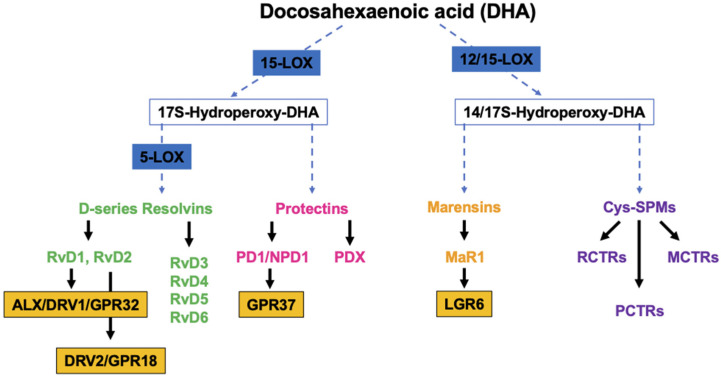
Overview of the pathways of the endogenous synthesis of specialized pro-resolving mediators (SPMs) from docosahexaenoic acid (DHA). LOX, lipoxygenase. Note that not all intermediates and enzymes are indicated.

**Figure 2 marinedrugs-22-00017-f002:**
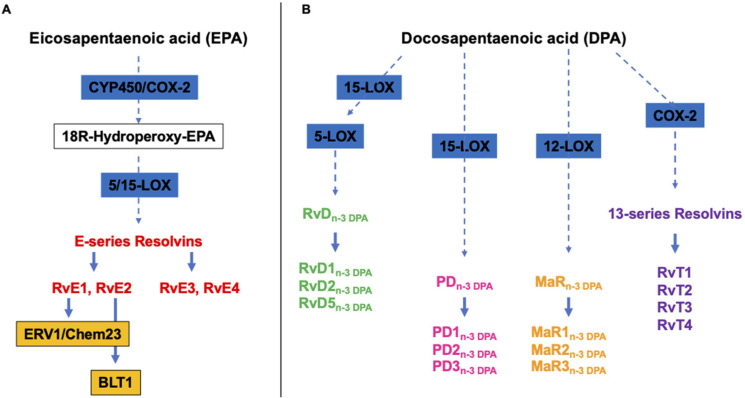
Overview of the pathways of the endogenous synthesis of specialized pro-resolving mediators (SPMs) from (**A**) eicosapentaenoic (EPA) and (**B**) docosapentaenoic (DHA) acid. CYP450, cytochrome 450 mixed function oxidase enzymes; COX, cyclooxygenase; LOX, lipoxygenase. It should be noted that not all intermediates and enzymes are indicated.

**Figure 3 marinedrugs-22-00017-f003:**
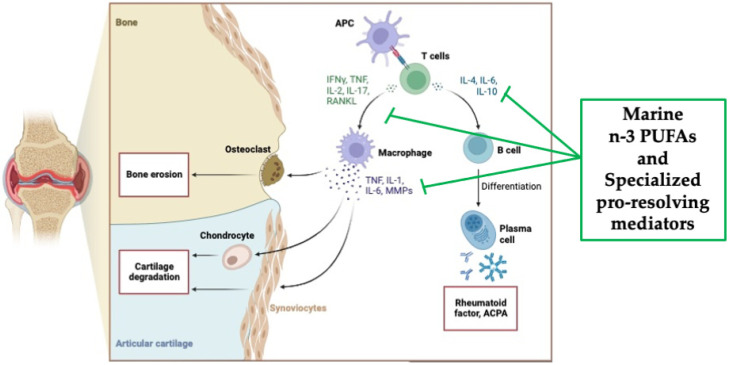
Overview of the pathological pathways in rheumatoid arthritis and the potential anti-inflammatory mechanisms of action of marine n-3 PUFAs as well as specialized pro-resolving mediators (SPMs) (created with Biorender.com, accessed on 22 November 2023).

## Data Availability

Not applicable.

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
