# Peer review of "The Role of Marine n-3 Polyunsaturated Fatty Acids in Inflammatory-Based Disease: The Case of Rheumatoid Arthritis"

_marinedrugs, 2023, doi:10.3390/md22010017_

Round 1
Reviewer 1 Report
Comments and Suggestions for Authors
Comments to the Authors of manuscript number: marinedrugs-2775104 entitled “The role of marine n-3 polyunsaturated fatty acids on inflammatory-based disease: the case of Rheumatoid Arthritis”.
Inflammation is a protective process involving immune and non-immune cells, but persistent inflammation can lead to chronic conditions. This article reviews the impact of marine n-3 polyunsaturated fatty acids (EPA and DHA) on inflammatory processes and their role in modulating human diseases like rheumatoid arthritis.
The review is very interesting and worth to publish.
1. introduction: The text provides an overview of chronic non-communicable diseases (NCDs) characterized by elevated inflammatory markers. It discusses the conserved inflammatory process and the crucial "resolution phase" involved in tissue repair. The importance of maintaining a balance between pro-inflammatory (M1) and anti-inflammatory (M2) macrophages is highlighted, along with the role of various cellular actors in the resolution of inflammation. The focus then shifts to "low-grade systemic chronic inflammation," contributing to tissue damage and increased susceptibility to NCDs. The latter part emphasizes the therapeutic approach to positively impact the "resolution phase" and introduces rheumatoid arthritis (RA) as an autoimmune disease associated with joint damage. The text suggests that dietary components, particularly omega-3 polyunsaturated fatty acids (n-3 PUFAs) from fish, may influence modifiable risk factors for chronic diseases. The article aims to review recent literature on the modulation of inflammatory-based diseases, such as RA, by n-3 PUFAs.
2. The text provides information about omega-3 polyunsaturated fatty acids (n-3 PUFAs) and specialized pro-resolving mediators (SPMs). It introduces omega-3 fatty acids, such as eicosapentaenoic acid (EPA) and docosahexaenoic acid (DHA), found in fish and supplements. The anti-inflammatory activity of n-3 PUFAs involves their incorporation into cell membranes, leading to decreased arachidonic acid (ARA) levels and reduced production of inflammatory eicosanoids. The text further explains the impact of n-3 PUFAs on inflammatory signaling pathways, including the nuclear factor kappa B (NFkB) pathway. The ability of n-3 PUFAs to interfere with NFkB activation is detailed, involving mechanisms like activation of peroxisome proliferator-activated receptor (PPAR)-gamma, interference with membrane raft formation, and binding to G-protein-coupled cell membrane receptor (GPR120). Additionally, the text highlights the role of n-3 PUFAs as substrates for the synthesis of specialized pro-resolving lipid mediators (SPMs), including resolvins (Rvs), protectins (PD1/NPD1), maresins (MaRs), and cysteinyl-SPMs (cys-SPMs). SPMs play a crucial role in inflammation resolution, limiting granulocyte chemotaxis, stimulating macrophage functions, and promoting tissue repair. The article concludes by mentioning studies demonstrating the therapeutic efficacy of SPMs in animal models of inflammation, suggesting a potential strategy for treating inflammatory-based pathologies.
3. L 203- anatomical term are needed
4. L 207 – the sentence seems to be not finished. Change?
5. L 238-240- it is synovitis
6. L 243- it is worth to mention about osteosarcopenia
7. L 251- mention it earlier
8. What about TGFβ/ALK/SMAD/MMP-13 or BMP signaling pathway?
9. L 283 – maybe it is worth to explain RANK/RANKL/OPG system
10. The text appears well-structured and informative, discussing the influence of modifiable factors, such as smoking, alcohol, and unsaturated fatty acids (FAs), on the development and progression of rheumatoid arthritis (RA). It emphasizes the importance of nutritional intervention, lifestyle changes, and popular nutritional approaches like the Mediterranean diet and dietary supplements, especially fish oil or n-3 PUFAs.
11. Ensure consistent usage of abbreviations. For example, use either "n-3 PUFAs" or "omega-3 PUFAs" consistently throughout the text.
12. The sentence "Two experimental studies showed that n-3 PUFAs, i.e. EPA and DHA, reduced the incidence and severity of arthritis in two different animal models of arthritis [95,96,20], with EPA being the more effective compared with DHA" could be rephrased for better clarity.
13. Break down complex sentences into shorter ones for better readability. For example, the sentence starting with "Of note, nutrition has been shown..." is quite long and could be divided into smaller, clearer sentences.
14. Ensure consistency in citing studies. For example, the sentence "A meta-analysis of 23 randomized placebo-controlled trials revealed only a trend toward benefit following n-3 PUFAs supplementation for improving RA disease parameters [119]" could benefit from specifying the meta-analysis being referred to.
15. In statements such as "The results of this meta-analysis suggest that n-3 PUFAs supplementation reduces patient assessed pain, morning stiffness, number of painful and/or tender joints, and NSAIDs consumption," provide more specific details about the magnitude of the effects and the level of significance if available.
Author Response
Ms. No.: MD_ 2775104
Title: The role of marine n-3 polyunsaturated fatty acids on inflammatory-based disease: the case of Rheumatoid Arthritis
Reviewer: 1
Comments to the Authors of manuscript number: marinedrugs-2775104 entitled “The role of marine n-3 polyunsaturated fatty acids on inflammatory-based disease: the case of Rheumatoid Arthritis”.
Inflammation is a protective process involving immune and non-immune cells, but persistent inflammation can lead to chronic conditions. This article reviews the impact of marine n-3 polyunsaturated fatty acids (EPA and DHA) on inflammatory processes and their role in modulating human diseases like rheumatoid arthritis.
The review is very interesting and worth to publish.
1. introduction: The text provides an overview of chronic non-communicable diseases (NCDs) characterized by elevated inflammatory markers. It discusses the conserved inflammatory process and the crucial "resolution phase" involved in tissue repair. The importance of maintaining a balance between pro-inflammatory (M1) and anti-inflammatory (M2) macrophages is highlighted, along with the role of various cellular actors in the resolution of inflammation. The focus then shifts to "low-grade systemic chronic inflammation," contributing to tissue damage and increased susceptibility to NCDs. The latter part emphasizes the therapeutic approach to positively impact the "resolution phase" and introduces rheumatoid arthritis (RA) as an autoimmune disease associated with joint damage. The text suggests that dietary components, particularly omega-3 polyunsaturated fatty acids (n-3 PUFAs) from fish, may influence modifiable risk factors for chronic diseases. The article aims to review recent literature on the modulation of inflammatory-based diseases, such as RA, by n-3 PUFAs.
2. The text provides information about omega-3 polyunsaturated fatty acids (n-3 PUFAs) and specialized pro-resolving mediators (SPMs). It introduces omega-3 fatty acids, such as eicosapentaenoic acid (EPA) and docosahexaenoic acid (DHA), found in fish and supplements. The anti-inflammatory activity of n-3 PUFAs involves their incorporation into cell membranes, leading to decreased arachidonic acid (ARA) levels and reduced production of inflammatory eicosanoids. The text further explains the impact of n-3 PUFAs on inflammatory signaling pathways, including the nuclear factor kappa B (NFkB) pathway. The ability of n-3 PUFAs to interfere with NFkB activation is detailed, involving mechanisms like activation of peroxisome proliferator-activated receptor (PPAR)-gamma, interference with membrane raft formation, and binding to G-protein-coupled cell membrane receptor (GPR120). Additionally, the text highlights the role of n-3 PUFAs as substrates for the synthesis of specialized pro-resolving lipid mediators (SPMs), including resolvins (Rvs), protectins (PD1/NPD1), maresins (MaRs), and cysteinyl-SPMs (cys-SPMs). SPMs play a crucial role in inflammation resolution, limiting granulocyte chemotaxis, stimulating macrophage functions, and promoting tissue repair. The article concludes by mentioning studies demonstrating the therapeutic efficacy of SPMs in animal models of inflammation, suggesting a potential strategy for treating inflammatory-based pathologies.
I thank the Reviewer for the careful revision and for the helpful comments. Below are the answers to all the points raised. Changes/additions to the original manuscript are written in red.
3. L 203- anatomical term are needed
A sentence is now present in the revised manuscript (please, see from line 213 to line 215). Thank you.
4. L 207 – the sentence seems to be not finished. Change?
I thank the Reviewer for having pointed out this lack in the sentence at line 207. I have change it, as you can see in the revised version of the manuscript from line 220 to line 223.
5. L 238-240- it is synovitis
Following the Reviewer’s suggestion, I have included this definition in the revised version of the manuscript at line 252. Thank you.
6. L 243- it is worth to mention about osteosarcopenia
I really thank the Reviewer for the need of this specification. A sentence on this regard is now present, from line 261 to line 262, of the revised manuscript.
7. L 251- mention it earlier
Following the Reviewer’s suggestion, we have moved this sentence at line 255 of the revised version of the manuscript. Thank you.
8. What about TGFβ/ALK/SMAD/MMP-13 or BMP signaling pathway?
I thank the reviewer for highlighted this lack in the RA paragraph. Following his/her indication, a sentence is present in the revised version of the manuscript from line 295 to line 299.
9. L 283 – maybe it is worth to explain RANK/RANKL/OPG system
Following the Reviewer’s suggestion, I have included a comment regarding this metabolic pathway (please, see from line 302 to line 306). Thank you.
10. The text appears well-structured and informative, discussing the influence of modifiable factors, such as smoking, alcohol, and unsaturated fatty acids (FAs), on the development and progression of rheumatoid arthritis (RA). It emphasizes the importance of nutritional intervention, lifestyle changes, and popular nutritional approaches like the Mediterranean diet and dietary supplements, especially fish oil or n-3 PUFAs.
11. Ensure consistent usage of abbreviations. For example, use either "n-3 PUFAs" or "omega-3 PUFAs" consistently throughout the text.
The Reviewer correctly pointed out this issue. In the revised version of the manuscript is now present only the terminology “n-3” (please, see lines 91 and 93). Thank you.
12. The sentence "Two experimental studies showed that n-3 PUFAs, i.e. EPA and DHA, reduced the incidence and severity of arthritis in two different animal models of arthritis [95,96,20], with EPA being the more effective compared with DHA" could be rephrased for better clarity.
Following the Reviewer’s suggestion, we have modified this sentence. Please, see from line 350 to line 351 of the revised manuscript. Thank you.
13. Break down complex sentences into shorter ones for better readability. For example, the sentence starting with "Of note, nutrition has been shown..." is quite long and could be divided into smaller, clearer sentences.
I thank the reviewer for this comment. I have modified the text accordingly to the Reviewer’s suggestion. Please, see starting from line 372 to the end of the paragraph in the revised version of the manuscript.
14. Ensure consistency in citing studies. For example, the sentence "A meta-analysis of 23 randomized placebo-controlled trials revealed only a trend toward benefit following n-3 PUFAs supplementation for improving RA disease parameters [119]" could benefit from specifying the meta-analysis being referred to.
I thank the Reviewer for this suggestion. In the revised version of the manuscript a new sentence is present (please, see from line 446 to line 449).
15. In statements such as "The results of this meta-analysis suggest that n-3 PUFAs supplementation reduces patient assessed pain, morning stiffness, number of painful and/or tender joints, and NSAIDs consumption," provide more specific details about the magnitude of the effects and the level of significance if available.
Following the Reviewer’s suggestion, I have added to the text the specific details (please, see from line 397 to line 403 of the revised manuscript). Thank you.

Reviewer 2 Report
Comments and Suggestions for Authors
The manuscript includes an interesting study. It is well presented and justified. However, before a new revision is done, performances ought to be carried out.
Abstract
Line 10: involves. Also in line 34.
Section 2
Both ω-3 and n-3 are currently accepted. The author ought to choose one of them for being employed throughout the manuscript. A mixed employment may confuse the reader.
Line 95: carbons.
Line 100: Maybe it could be mentioned that ARA belongs to the ω6 series.
Some comment on the health relevance of the ω3/ ω6 ratio and also of the DHA/EPA ratio could be included.
Section 3
As being the main section of the manuscript, some comment on the analytical tools employed could be included. For me, this is the main lack of the review, i.e., the explanation of the analytical tools employed in previous studies.
Conclusions
The author could mention, at least in this section, the need for advanced analytical tools such as lipidomics and proteomics. Such tools would be mandatory to elucidate the preservative mechanisms.
Comments on the Quality of English Language
Minor performances could be done.
Author Response
Ms. No.: MD_ 2775104
Title: The role of marine n-3 polyunsaturated fatty acids on inflammatory-based disease: the case of Rheumatoid Arthritis
Reviewer #2:
The manuscript includes an interesting study. It is well presented and justified. However, before a new revision is done, performances ought to be carried out.
I thank the Reviewer for the careful revision and for the helpful comments. Below are the answers to all the points raised. Changes/additions to the original manuscript are written in red.
Abstract
Line 10: involves. Also in line 34.
I really thank the Reviewer for having identified these grammatical mistakes. The corrections have been performed, see at lines 9 and 33.
Section 2
Both ω-3 and n-3 are currently accepted. The author ought to choose one of them for being employed throughout the manuscript. A mixed employment may confuse the reader.
The Reviewer correctly pointed out this issue. In the revised version of the manuscript is now present only the terminology n-3 (please, see lines 91 and 93). Thank you.
Line 95: carbons.
Again, I really thank the Reviewer for having identified this grammatical mistake. The correction has been performed, see at line 94.
Line 100: Maybe it could be mentioned that ARA belongs to the ω6 series.
Following the Reviewer’s suggestion, I have added a sentence on this regard (please, see from line 105 to line 106). Thank you.
Some comment on the health relevance of the ω3/ ω6 ratio and also of the DHA/EPA ratio could be included.
I thank the Reviewer for having highlighted this point. Comments on this regard are present in the revised version of the manuscript (please, see from line 110 to line 120).
Section 3
As being the main section of the manuscript, some comment on the analytical tools employed could be included. For me, this is the main lack of the review, i.e., the explanation of the analytical tools employed in previous studies.
The Reviewer correctly pointed out this gap in the paragraph 3 “Rheumatoid Arthritis (RA). Following his/her suggestion a comment is now present in the revised manuscript. Please, see from line 321 to line 328.
Conclusions
The author could mention, at least in this section, the need for advanced analytical tools such as lipidomics and proteomics. Such tools would be mandatory to elucidate the preservative mechanisms.
Following the Reviewer’s suggestion, I have added this important new issue (please, see from line 467 to line 474. Thank you.
Comments on the Quality of English Language
Minor performances could be done.
I thank the Reviewer for this observation. I went throughout the text to improve the quality of the English Language.

Round 2
Reviewer 2 Report
Comments and Suggestions for Authors
The manuscript has been performed according to previous comments. I woudl recommend its acceptation.
Comments on the Quality of English Language
Minor editing could be done.